# Renal Function Parameters in Distinctive Molecular Subtypes of Prostate Cancer

**DOI:** 10.3390/cancers15205013

**Published:** 2023-10-16

**Authors:** Andrei Daniel Timofte, Irina-Draga Caruntu, Adrian C. Covic, Monica Hancianu, Nona Girlescu, Mariana Bianca Chifu, Simona Eliza Giusca

**Affiliations:** 1Department of Morpho-Functional Sciences I, “Grigore T. Popa” University of Medicine and Pharmacy, 700115 Iasi, Romania; nona.girlescu@yahoo.com (N.G.); bianca.manole@ymail.com (M.B.C.); simonaelizagiusca@gmail.com (S.E.G.); 2Department of Pathology, “Dr. C. I. Parhon” University Hospital, 700503 Iasi, Romania; 3Romanian Medical Science Academy, 030171 Bucharest, Romania; accovic@gmail.com; 4Romanian Academy of Scientists, 50044 Bucharest, Romania; 5Department Medical II, “Grigore T. Popa” University of Medicine and Pharmacy, 700115 Iasi, Romania; 6Department of Nephrology, Dialysis and Renal Transplant Center, “Dr. C. I. Parhon” University Hospital, 700503 Iasi, Romania; 7Department of Pharmaceutical Sciences II, “Grigore T. Popa” University of Medicine and Pharmacy, 700115 Iasi, Romania; monica.hancianu@umfiasi.ro

**Keywords:** prostate cancer, renal function, molecular subtypes, ERG, SPINK1, HOXB13, TFF3

## Abstract

**Simple Summary:**

Prostate cancer is a malignancy with varying clinical outcomes, and monitoring renal function is crucial for patient care. Our study investigates the relationship between renal function and distinct molecular subtypes of prostate adenocarcinomas. We analyzed 72 patients with prostate cancer and chronic kidney disease who underwent radical prostatectomy. We categorized patients based on molecular markers and found that the ERG+/SPINK1+ subgroup had higher postoperative kidney disease stages and serum creatinine levels compared to the ERG+/SPINK1− subgroup, suggesting a connection between SPINK1 overexpression and kidney function. The HOXB13 −/TFF3+ subgroup exhibited higher preoperative serum creatinine levels and kidney disease stages than the HOXB13−/TFF3− subgroup, implying a role for TFF3 in kidney function. Furthermore, our study revealed links between kidney disease stages and prognostic grade groups in different molecular subtypes, highlighting the complex interplay between kidney function and tumor behavior. Our research underscores the importance of considering molecular subtypes in prostate cancer management.

**Abstract:**

Prostate cancer is a prevalent malignancy in male patients, having diverse clinical outcomes. The follow-up of patients diagnosed with prostate cancer involves the evaluation of renal function, because its impairment reduces patient survival rates and adds complexity to their treatment and clinical care. This study aimed to investigate the relationship between renal function parameters and distinctive molecular subtypes of prostate adenocarcinomas, defined by the immunoexpression of the SPINK1, ERG, HOXB13, and TFF3 markers. The study group comprised 72 patients with prostate cancer and associated chronic kidney disease (CKD) who underwent radical prostatectomy. Histopathological, molecular, and renal parameters were analyzed. Patients were categorized based on ERG/SPINK1 and HOXB13/TFF3 status, and correlations with renal function and prognostic grade groups were assessed. The ERG+/SPINK1+ subgroup exhibited significantly higher postoperative CKD stages and serum creatinine levels compared to the ERG+/SPINK1− subgroup. This suggests an intricate relationship between SPINK1 overexpression and renal function dynamics. The HOXB13−/TFF3+ subgroup displayed higher preoperative serum creatinine levels and CKD stages than the HOXB13−/TFF3− subgroup, aligning with TFF3’s potential role in renal function. Furthermore, the study revealed associations between CKD stages and prognostic grade groups in different molecular subtypes, pointing out an intricate interplay between renal function and tumor behavior. Although the molecular classification of prostate acinar ADK is not yet implemented, this research underscores the variability of renal function parameters in different molecular subtypes, offering potential insights into patient prognosis.

## 1. Introduction

Prostate cancer is the second most common male neoplasia, affecting 1 in 7 men worldwide and accounting for 4% of cancer-associated deaths [1]. The main histological type of prostate tumors is acinar adenocarcinomas (ADK)—conventional form, followed by other histological variants (95%), and only a small percentage (5%) is classified as non-acinar carcinoma [2]. For cases diagnosed at early stages, the 5-year survival reaches 100%, while for advanced, metastatic stages, the 5-year survival drops below 30% [1].

The diagnosis of prostate cancer has been based, since 1966, on the Gleason score and has been refined through successive modifications, translated in 2016 into the prognostic grade groups established by the International Society of Urological Pathology [3]. Progress in the diagnosis and classification of prostate cancer has led to a classification framework that ensures not only reproducibility but also a more precise prognostic assessment. Nowadays, the pathologists’ efforts are directed towards combining the microscopical aspects with the molecular and genetic background. Thus, criteria for a molecular classification are highly discussed, opening new perspectives for a personalized therapy [4,5,6].

As life expectancy continues to rise, and advancements in screening, diagnosis, and treatment contribute to a decline in neoplastic-specific mortality, there is a concerning increase in mortality rates attributed to non-neoplastic causes, such as cardiovascular, diabetes, and renal conditions [7,8]. Among these, cardiovascular disease emerges as the second most prominent cause of death in prostate cancer patients [9]. Consequently, several studies have increasingly focused on understanding the associated risk factors and prevalence of cardiovascular disease, particularly in patients undergoing androgen therapy for prostate cancer [10,11,12]. It has been postulated that testosterone suppression may directly impact the vascular endothelium, promoting adverse cardio-metabolic factors like inflammation, dyslipidemia, insulin resistance, and atherosclerosis [13,14]. Moreover, recent research on the relation between the biologically natural, unaltered progression of prostate neoplasia and its possible implications in cardiovascular disease delves into the molecular connections between these two entities, revealing that several genes (such as DES, ACTC1, OR51E2, CACNA1D, TBX18, PLN, and CASQ2) responsible for maintaining a proper cardiovascular system function are downregulated in prostate cancer patients [15].

In a similar manner, a pressing concern in the management of prostate cancer patients is the assessment of chronic kidney disease (CKD) and acute kidney injury incidence. The coexistence of impaired renal function and prostate neoplasia not only reduces patient survival rates but also complicates treatment and clinical care. Recent findings confirm that the concurrent use of androgen deprivation therapy and radiotherapy is associated with an elevated risk of acute renal failure in patients diagnosed with prostate ADK [16]. Specifically, androgen deprivation therapy has a direct adverse impact on renal blood vessels by reducing testosterone levels, and induces estrogen deficiency, which is associated with negative effects on renal tubular function, and also determines hyperglycemia and dyslipidemia with potential repercussion on the integrity of normal tubular interstitial membranes [17,18,19].

To the best of our knowledge, few studies have investigated renal dysfunction in ADK patients without androgen deprivation therapy [20,21]. One study analyzed multiple renal function parameters (namely, urea, creatinine, and cystatin C) in untreated versus treated prostate cancer patients versus healthy subjects, showing only increased cystatin C levels in hormonally-treated ADK patients compared to untreated ones, and significantly higher levels of serum urea and creatinine coupled with reduced cystatin C levels in prostate cancer patients (treated and untreated) compared to healthy subjects [20]. The other study examined preoperative cysteine C levels in benign prostate hyperplasia, prostatic intraepithelial neoplasia, and hormonally-untreated prostate cancer patients, concluding that this parameter alone is not a reliable predictor for these pathological conditions [21].

Translating the classical histopathological diagnosis of ADK to a new framework that defines molecular subtypes may lead to the validation of potential indicators of clinical course. Within this context, renal function analysis can be integrated, as a risk element and potential indicator of the degree of aggressiveness of prostate acinar ADK, allowing a better stratification of patients, in accordance with the molecular classification framework [22].

Based on our previous results that defined particular molecular subtypes of prostate acinar ADK [23], the present research aims to study renal function as a potential factor for aggressive malignancy course, and its implication in prognostic assessment.

## 2. Materials and Methods

### 2.1. Patients

We conducted an observational and correlational study on a group consisting of 72 patients with prostate ADK and associated CKD who underwent radical prostatectomy within the “Dr. C. I. Parhon” University Hospital, Iaşi, Romania, between 2010 and 2018. The research received the approval of the Research Ethics Committee of the “Grigore T. Popa” University of Medicine and Pharmacy, Iaşi (No. 292/2023).

The inclusion and exclusion criteria and the design of the study are summarized in Figure 1.

### 2.2. Pathological Diagnosis and Renal Function Parameters

Tissue samples were collected from radical prostatectomy specimens immediately post-surgery and fixed in 10% formalin solution. These samples followed the standard histopathological examination protocol: dehydration (using increasing concentrations of alcohol from 50% to 100%), clearing with xylene, and paraffin embedding. Paraffin blocks were cut into 4–5 micron-thick sections and placed on microscopic slides, which were consequently deparaffinated and stained in hematoxylin and eosin (H&E). For all cases, the corresponding slides stained in H&E were reevaluated microscopically in accordance to the latest edition of the World Health Organization (WHO) classification for urinary and male genital tumors published in 2022 [2,24], re-assessing the Gleason score, dominant pattern, and aggressiveness parameters (capsular, perineural, and lymphovascular invasion, pathological tumor-node-metastasis—pTNM stage) and assigning prognostic grade groups.

The renal function parameters (serum creatinine and serum urea) before and after prostatectomy were also documented from the patients’ medical records. Based on serum creatinine levels and age, estimated preoperative and postoperative GFRs were calculated using the Chronic Kidney Disease Epidemiology Collaboration (CKD-EPI) 2021 calculation formula, and patients were classified into different stages of renal dysfunction according to the KDIGO 2012 guidelines [25,26].

### 2.3. Immunohistochemical Analysis

Immunohistochemical (IHC) analysis was conducted on paraffin-embedded prostatic tissue blocks originating from radical prostatectomy procedures, previously used for histopathological diagnosis, according to the datasheet provided by the antibody manufacturer [27,28]. Sequential 4 μm-thick tissue sections were affixed onto adhesive positively charged slides, with the entire process executed manually. Initially, the slides were deparaffinized in two xylene baths, followed by rehydration through a series of decreasing alcohol concentrations (100%, 90%, 80%, and 70%), and then rinsed with distilled water. The antigen was unveiled through heat-induced epitope retrieval (HIER), immersing the slides in a tris-ethylenediaminetetraacetate (EDTA) solution (pH 9), and subsequently, heating at 97 °C for 25 min. Post-HIER, the slides were rinsed in distilled water, cooled to room temperature, and endogenous peroxidases were neutralized with hydrogen peroxide for 10 min.

Subsequent to the application of the ERG, SPINK1, HOXB13, and TFF3 antibodies (Table 1), the slides were incubated overnight at 4 °C. Both streptavidin peroxidase and the secondary antibody (goat anti-rabbit IgG ab97051, Abcam, Cambridge, MA, USA) were applied for 30 min at room temperature. Following each incubation step, the slides were washed with phosphate-buffered saline (PBS) solution for 5 min. To prompt the immunological response, a solution of 3,3′-Diamino-Benzidine (DAB) chromogen was employed, subsequently followed by Mayer’s hematoxylin counterstaining.

Positive and negative external and internal controls were performed simultaneously with the IHC reactions for the investigated markers [Appendix A.

The internal positive control for ERG featured nuclear immunoreaction in endothelial cells, and the positive control for SPINK1 was established through the cytoplasmic immunoreaction in pancreatic acinar cells [29,30]. The validation of HOXB13 as a positive control involved observing nuclear immunoreaction in the epithelial cells of benign prostate glands [31]. To confirm TFF3’s positive control, we examined the cytoplasmic immunoreaction in colonic goblet cells [32].

The negative control for HOXB13 was represented by slides with human brain tissue—a HOXB13-negative tissue; for the rest of the antibodies, the incubation of prostatic ADK tissue sections with the primary antibody was excluded [27,28,31].

### 2.4. Scoring System

The semi-quantitative assessment of the antibodies’ immunoexpression was performed, applying adapted scoring systems from previous research related to these markers (Table 2).

### 2.5. Statistical Analysis

For the statistical analysis, Statistica version 7 (Tibco, Palo Alto, CA, USA) and Excel 2016 version 16.0 (Microsoft, Redmond, WA, USA) were used.

Comparisons between the mean values of the variables included in the study for each defined variant of the immunohistochemical profile were performed with the *t*-test for independent variables, at the critical significance level *p* = 0.05.

Comparisons between the defined immunohistochemical variants on preoperative CKD stages’ associations with the prognostic grade groups were performed with the Chi-square test at the critical significance level *p* = 0.05, with Yates correction (given the small dimensions of the subgroups). A Spearman R test was used to determine the overall correlation between the preoperative CKD stages and the prognostic grade groups, at a significance level of *p* = 0.05.

## 3. Results

### 3.1. Clinico-Pathological Characteristics

The main clinico-pathological characteristics of the study group, after microscopical reevaluation, are presented in Table 3.

Microscopic aspects illustrating Gleason patterns transposed into the Gleason scoring system relevant for the histopathological diagnosis are shown in Figure 2.

Regarding the histopathological variants, all cases were conventional acinar ADK, some of them presenting an associated variant: foamy cell (9 cases—12.5%), mucinous (5 cases—6.94%), atrophic and microcystic (one case each—0.72%). Patients were comprised into prognostic grade groups according to their Gleason scores as follows: 17 (23.61%) patients in grade Group 1 with a Gleason score of 6, 38 (52.77%) patients in grade Group 2 with a Gleason score of 7 (=3 + 4), 5 (6.94%) patients in grade Group 3 with a Gleason score of 7 (=4 + 3), 6 (8.33%) patients in grade Group 4, and 6 (8.33%) patients in grade Group 5. Gleason Pattern 3 predominated in 51 cases (76.38%), followed by Pattern 4 in 16 cases (22.22%), and Pattern 5 in 1 case (1.38%).

Invasion of the prostate capsule, both intra- and extracapsular, was evident in 66 cases (91.66%) but not in the remaining 6 (8.33%). While tumor emboli in the lympho-vascular spaces were observed in 15 (20.83%) cases, without evidence in the other 57 (79.16%) cases, perineural invasion was present in 61 (84.72%) cases, lacking in 11 (15.27%) cases. In the majority of cases, the tumor was confined to the prostate—pT2 stage (53 cases—73.61%); in 19 cases (26.38%), the tumoral process extended outside the prostate presenting a pT3 stage.

The postoperative PSA serum levels confirmed that almost half of the patients (35 cases—48.61%) had biochemical recurrence (PSA ≥ 0.2 ng/mL).

### 3.2. Immunohistochemical Profile

ERG immunoexpression was relatively homogeneous in all labelled tumor cells, of moderate to high intensity, and comparable to that of the internal positive labelling (endothelial cell nuclei). The evaluation of ERG immunoexpression revealed 40 cases (55.55%) with a positive ERG status and 32 cases (44.44%) with a negative ERG status. Cases with a positive ERG status showed nuclear positivity in more than 90% of tumor cells. In benign prostate tissue, ERG expression was completely absent.

SPINK1 immunostaining was characterized by heterogeneity of the cytoplasmic expression of tumor cells; of the 72 prostate ADK cases analyzed, 61 cases (84.72%) were classified as having a SPINK1 negative status and 11 cases (15.27%) had a SPINK1 positive status.

For all 72 cases, HOXB13 immunostaining in tumor glands, assessed in comparison to the intensity of luminal secretory cells of benign adjacent prostatic glands, was characterized by intra- and inter-tumor heterogeneity. The low HOXB13 immunoexpression subgroup comprised 47 (65.27%) cases and the high HOXB13 immunoexpression subgroup included 25 (34.72%) cases.

TFF3 immunostaining, characterized by a granular, “dot-like” cytoplasmic labelling, revealed a low immunoexpression in 41 (56.94%) cases and a high immunoexpression in 31 (43.05%) cases.

Relevant aspects for the tissular immunoexpression pattern of the markers used in our study are illustrated in Figure 3.

By analyzing ERG and SPINK1 co-expression, four different subgroups were defined, as follows: Subgroup 1, with positive immunoexpression of both ERG and SPINK1 (E+S+)—4 (5.55%) cases; Subgroup 2, with a positive ERG status and negative SPINK1 immunoexpression (E+S−)—36 (50%) cases; Subgroup 3, with negative ERG immunoexpression and positive SPINK1 immunoexpression (E−S+)—9 (12.5%) cases; Subgroup 4, with both ERG and SPINK1 negative immunoexpression (E−S−)—23 (31.94%) cases.

Also, HOXB13 and TFF3 co-expression led to the separation in four different subgroups, as follows: Subgroup 1, with high immunoexpression of HOXB13 and TFF3 (H+T+)—22 (30.55%) cases; Subgroup 2, with high HOXB13 immunoexpression and low TFF3 immunoexpression (H+T−)—3 (4.11%) cases; Subgroup 3, with low HOXB13 immunoexpression and high TFF3 immunoexpression (H−T+)—9 (12.5%) cases; Subgroup 4, with both HOXB13 and TFF3 low immunoexpression (H−T−)—38 (52.77%) cases.

### 3.3. Renal Parameters

In regard to the preoperative renal parameters, serum creatinine levels ranged from 0.6 to 2.2 mg/dL, serum urea from 11 to 62 mg/dL, and eGFR from 33 to 111 mL/min/1.73 m, the predominant CKD stage being Stage 1 (43 cases—59.72%), followed by Stage 2 and 3, respectively (21 cases—29.17% and 8 cases—11.11%).

After the radical prostatectomy, the renal parameters varied as follows: serum creatinine levels ranged from 0.67 to 1.7 mg/dL, serum urea from 17 to 66 mg/dL, and eGFR from 44 to 110 mL/min/1.73 m, the predominant CKD stage being Stage 1 (45 cases—62.5%), followed by CKD Stages 2 and 3, respectively (22 cases—30.55% and 5 cases—6.94%).

### 3.4. Correlations between Preoperative/Postoperative Renal Parameters in Subsets of Prostate Acinar ADK Defined by ERG/SPINK1 Status

The comparisons between the mean values of the preoperative and postoperative renal parameters, respectively, for each ERG/SPINK1 subgroup, are comprised in Table 4 and Table 5. No statistically significant differences were registered for preoperative serum creatinine, serum urea, eGFR, and CKD stages, when comparing ERG/SPINK1 subsets among themselves (Table 4). Regarding postoperative parameters, the E+S+ subgroup compared to the E−S+ subgroup had a significantly higher mean of serum creatinine levels (*p* = 0.02) and CKD stage (*p* = 0.04) (Table 5). Also, we noted a significantly higher postoperative CKD stage in the E+S+ subgroup when compared to the E+S− subgroup (*p* = 0.03) (Table 5).

We found no statistically significant differences between preoperative and postoperative levels of all renal parameters, according to the IHC profile defined by ERG and SPINK1 markers (*p* > 0.05).

### 3.5. Correlations between Preoperative/Postoperative Renal Parameters in Subsets of Prostate Acinar ADK Defined by HOXB13/TFF3 Status

The comparisons between the mean values of the preoperative and postoperative renal parameters, respectively, for each HOXB13/TFF3 subgroup, are comprised in Table 6 and Table 7. Regarding the preoperative parameters, the H−T+ subgroup compared to the H−T− subgroup had a significantly higher mean of serum creatinine levels (*p* = 0.01) and CKD stage (*p* = 0.01); we also noted a significantly lower mean of eGFR in the H−T+ subgroup when compared to the H−T− subgroup (*p* = 0.02) (Table 6).

No statistically significant differences were registered for postoperative serum creatinine, serum urea, eGFR, and CKD stages, when comparing HOXB13/TFF3 subsets among themselves (Table 7).

We found no statistically significant differences between preoperative and postoperative levels of all renal parameters, according to the IHC profile defined by HOXB13 and TFF3 markers (*p* > 0.05).

### 3.6. Correlations between Preoperative/Postoperative Renal Parameters and Aggressive Histopathological Features in Subsets of Prostate Acinar ADK Defined by ERG/SPINK1 and HOXB13/TFF3 Status, Respectively

The comparisons of the mean values for the preoperative and postoperative renal parameters, respectively, in cases with aggressive behavior (considering capsular, perineural, and lymphovascular invasion) versus cases without these features, according to ERG/SPINK1 and HOXB13/TFF3 status, respectively, revealed no statistically significant correlations (*p* > 0.05).

### 3.7. Correlations between Preoperative CKD Stages and Prognostic Grade Groups

The comparisons between the different subgroups of prostate acinar ADK according to ERG/SPINK1 status, regarding the association of preoperative CKD stage with the prognostic grade groups, revealed significantly more CKD Stage 1 cases with prognostic grade Group 2 in the E+S− subgroup as compared to the E−S+ subgroup (*p* = 0.03) (Table 8).

On the other hand, similar comparisons between all subgroups of prostate acinar ADK defined by HOXB13/TFF3 immunoexpression indicated the following associations (Table 9):
-significantly more cases with CKD Stage 1 and prognostic grade Group 2 in the H−T− subgroup, as compared to the H+T+ subgroup (*p* = 0.046);-significantly more cases with CKD Stage 1 and prognostic grade Group 4 in the H+T+ subgroup versus the H−T− subgroup (*p* = 0.036);-significantly more cases with CKD Stage 1 and prognostic Grade Group 5 in the H+T3+ subgroup, as compared to the H−T− subgroup (*p* = 0.0097);-significantly more cases with CKD Stage 2 and prognostic grade Group 2 in the H−T+ subgroup, as compared to the H+T+ subgroup (*p* = 0.0057);-significantly more cases with CKD Stage 2 and prognostic grade Group 2 in the H−T+ subgroup, as compared to the H−T− subgroup (*p* = 0.009);-significantly more cases with CKD Stage 3 and prognostic grade Group 2 in the H−T+ subgroup as compared to the H−T− subgroup (*p* = 0.04).

Overall, calculating the Spearman R correlation coefficient, our results showed there is a small but significant inverse correlation between the prognostic grade groups and preoperative CKD stages (R = −0.2656, t(N−2) = 2.31, *p* = 0.02) (Figure 4).

## 4. Discussion

In the preoperative evaluation of prostate cancer patients, assessing renal function through standard parameters (serum creatinine, serum urea, and eGFR) holds significant clinical importance, playing an essential role in surgical decision-making, perioperative management, and patient outcomes.

The pathological features of the study group are consistent with the general profile of acinar prostate ADK reported in the literature, with conventional variant and prognostic grade Group 2 (Gleason score 7 = 3 + 4) being predominant [37]. In addition to the usual diagnostic approach, we defined two diagnostic classes by the ERG/SPINK1 and HOXB13/TFF3 profile, respectively, each class comprising four possible subgroups according to the low and high expression of each marker.

The novelty of our work lies in the comprehensive integration of both preoperative and postoperative renal function parameters with these molecular subtypes of prostate ADK. This research aligns with the current trend of validation of a molecular framework aiming to refine the diagnostic of prostate ADK. Moreover, our results point out the different behavior of molecular types not only in the prostatic site but also at the systemic level—including renal damage. Thus, our data add value for the study of kidney parameters in prostate ADK, due to the original approach focused on patients with renal function impairment developed during the natural course of this malignancy, and unaffected by androgen deprivation therapy. Furthermore, it is the only study, at the present moment, to analyze the relation of the prognostic grade groups of prostate ADK with the preoperative stages of CKD defined by eGFR.

Nowadays, researchers’ attention is focused on changes in renal function in the context of anti-androgen therapy [38,39]. The literature provides limited data on the prognostic value of individual parameters of renal function in prostatic neoplasia at the time of diagnosis. A recent study has examined the relationship between serum creatinine levels and prognostic grade groups in prostate cancer, highlighting that either low or high serum creatinine levels indicate a poor prognosis [40].

ERG and SPINK1 play an important part in prostate tumorigenesis [41,42,43,44]. Prostate cancer patients can exhibit distinctive ERG gene fusions, resulting in the upregulation of ERG expression [42]. SPINK1 overexpression was associated with an aggressive phenotype, with rapid clinical progression [45]. The SPINK1–ERG subgroups defined in our study align to the controversial data on the coexpression of these two markers in prostate cancer, supporting the theory that they may not exhibit absolute mutual exclusivity [43,46].

We demonstrated that the renal function, evaluated after radical prostatectomy throughout serum creatinine levels, was significantly affected in the E+S+ subgroup as compared to the E−S+ subgroup, suggesting that cases with ERG overexpression tend to have a negative clinical impact on kidney function. A possible explanation for this finding could reside in the role played by ERG on the vascular and hematopoietic system, serving as a distinct marker for the integrity of vascular endothelial cells, ensuring the stability of blood vessels [47].

Thus, ERG gene dysfunction characterizing prostate acinar ADK with ERG overexpression may act on glomerular capillaries responsible for renal filtration, thus explaining the increased creatinine levels and progression towards a higher CKD stage.

Meanwhile, the higher postoperative CKD stage in patients with E+S+ status as compared to E+S− and E−S+, respectively, can support the potential combined contribution of these molecules in affecting renal function, possibly through the ERG effects presented above, overlapping with SPINK1 involvement in renal MMP12 activation and determining the damage of the interstitial matrix and glomerular basement membrane [48,49].

HOXB13 and TFF3 are emerging markers with confirmed involvement in the complex process of prostate tumorigenesis [36,50]. However, the precise role of HOXB13 in prostate carcinogenesis still remains a matter of debate, as it is considered both an oncogene and a tumor suppressor gene, with direct implication in modulating androgen responsiveness [51,52,53]. On the other hand, the possible mechanism by which TFF3 promotes prostate tumor progression is not fully understood [36].

Our results revealing a statistically significant higher mean of preoperative serum creatinine, and consequently a higher mean of preoperative CKD stage in the H−T+ subgroup as compared to the H−T− subgroup are concordant with data that support the role of TFF3 as a marker of the future risk of CKD [54]. Ectopic synthesis of TFF3 was found primarily in proximal and distal tubules, and collecting ducts, with a possible role in the repair of tubular epithelium injury—high TFF3 urinary levels being indicative of ongoing damage and inflammatory processes [55,56].

Furthermore, our results showing a lower average preoperative eGFR in the H−T+ subgroup compared to the H−T− subgroup are in accordance with other studies reporting that serum or urinary levels of TFF3 are negatively correlated with creatinine clearance or eGFR [55,57]. These findings offer new perspectives for the use of TFF3 as a biomarker, not only for subtyping prostate cancer but also for the assessment of renal impairment among patients with prostate cancer characterized by TFF3 positivity.

Our study also demonstrated a higher frequency of preoperative CKD Stages 2 and 3 in patients with prognostic grade Group 2 in the H−T+ subgroup as compared to all other subgroups, suggesting an associated renal function damage in this molecular subtype, even at a Gleason score of 7 (3 + 4).

All in all, the kidney function seems to be more affected in prostate cancers with lower prognostic grade groups rather than in cases with higher grade groups, an observation sustained by a small but significant inverse correlation between the prognostic grade groups and preoperative CKD stages. Concretely, when CKD stages increase, prognostic grade groups show a decreasing trend, and conversely, when prognostic grade groups increase, CKD stages decrease. In our opinion, a possible explanation for this finding could be the fact that the lower the prognostic grade group, the slower the evolution of malignancy, thus the patients associate the age-related deterioration of renal function with the development of advanced CKD stages. These correlations warrant further exploration, potentially indicating a complex interplay between renal function and cancer aggressiveness.

Despite the originality of stratifying prostate ADK cases into molecular subgroups, and correlating them with renal function parameters, our study has some limitations. The first one is the relatively small size of the study group, which impacted the structure of the molecular subgroups. Due to the limited number of cases within certain subgroups, the generalizability of our findings may be constrained. Another limitation is that our analysis primarily focuses on renal function prior to and immediately after prostatectomy. While this provides valuable insights into the preoperative and early postoperative period, it does not encompass the long-term effects or changes that may occur over time, particularly in response to androgenic therapy. Therefore, this study can be considered a preliminary stage in our research. We acknowledge the need to expand our investigation by evaluating renal function in response to androgenic therapy in different molecular subgroups, allowing for a more comprehensive understanding on renal health in prostate cancer patients.

## 5. Conclusions

Our study sheds light on how the ERG, SPINK1, HOXB13, and TFF3 molecular markers may impact renal function, potentially influencing treatment decisions and patient outcomes. Specifically, ERG overexpression appears to negatively impact renal function, and combined ERG and SPINK1 effects may further contribute to renal impairment. TFF3 shows promise as a biomarker for both subtyping prostate cancer and assessing renal damage in TFF3-positive cases.

## Figures and Tables

**Figure 1 cancers-15-05013-f001:**
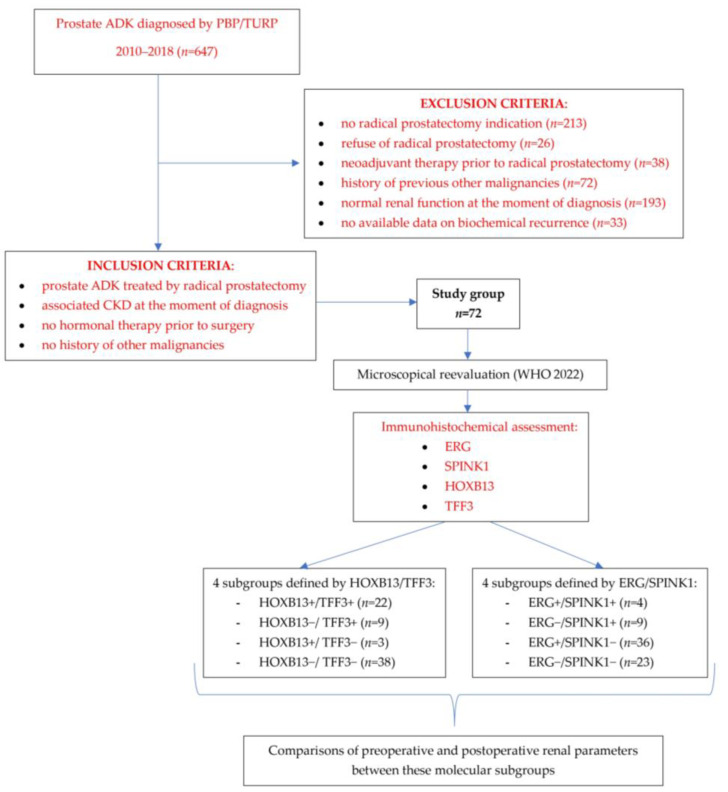
Flowchart to illustrate the study design.

**Figure 2 cancers-15-05013-f002:**
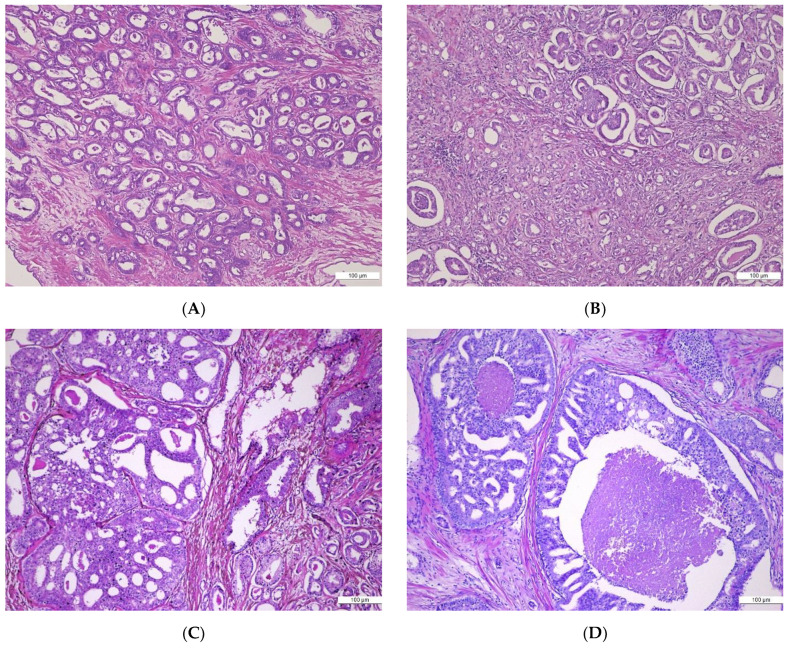
(**A**–**D**) Histological patterns in prostate acinar adenocarcinomas (ADK): (**A**) Prostate acinar ADK, Gleason Pattern 3, invasive up to the prostatic capsule’s extent, without surpassing it. H&E, ×10; (**B**) Prostate acinar ADK, Gleason Pattern 3 associating Gleason Pattern 4 (fused tumor glands, glomeruloid component), H&E, ×10; (**C**) Prostate acinar ADK, Gleason Pattern 4 cribriform and Gleason Pattern 3, H&E, ×10; (**D**) Prostate acinar ADK, Gleason Pattern 5—tumoral glands with central comedonecrosis, H&E.

**Figure 3 cancers-15-05013-f003:**
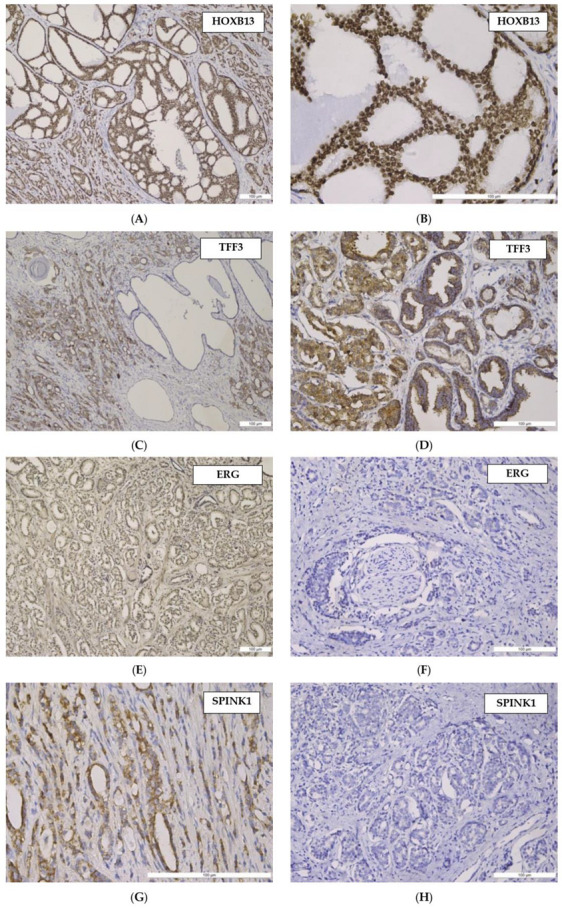
(**A**–**H**) Histological pattern and tissular immunoexpression of studied markers in prostate acinar adenocarcinomas (PAC): (**A**) HOXB13—moderate nuclear immunostaining in PAC, Gleason Pattern 3 and cribriform variant (Gleason Pattern 4). Anti- HOXB13 antibody immunostaining, ×100; (**B**) HOXB13—high nuclear immunostaining in PAC, Gleason Pattern 3, cribriform variant. Anti-HOXB13 antibody immunostaining, ×200; (**C**) TFF3—low and moderate cytoplasmic immunostaining in PAC, Gleason Pattern 3 and 4, negative staining in dilated normal prostate glands. Anti-TFF3 antibody immunostaining, ×100; (**D**) TFF3—moderate cytoplasmic immunostaining in PAC, Gleason Pattern 3, weak staining in normal prostate glands. Anti-TFF3 antibody immunostaining, ×100; (**E**) ERG—moderate nuclear immunostaining in PAC, in 90% of tumor cells—Gleason Pattern 3. Anti-ERG antibody immunostaining, ×100; (**F**) ERG—absence of nuclear immunostaining in PAC, perineural invasion. Anti-ERG antibody immunostaining, ×100; (**G**) SPINK1—moderate cytoplasmatic immunostaining in PAC. Anti-SPINK1 antibody immunostaining, ×200; (**H**) SPINK1—absence of cytoplasmatic immunostaining in PAC. Anti-SPINK1 antibody immunostaining, ×100.

**Figure 4 cancers-15-05013-f004:**
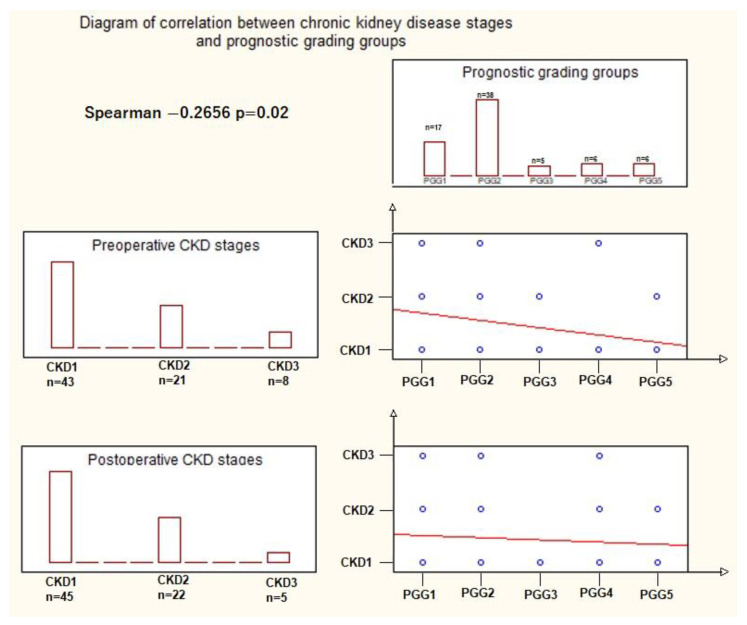
Correlation between the prognostic grade group (PGG) and preoperative chronic kidney disease (CKD) stage: the descending red line in the preoperative CKD/PGG correlation graphic indicates an inverse correlation between the two variables, where an increase in CKD class corresponds to a decrease in the PGG, and vice versa. The steepness of the red line reflects the statistically significant correlations, as opposed to the red line in the postoperative CKD/PGG correlation graphic. The blue dots signify specific associations between CKD class (1, 2, 3) and PGG (1, 2, 3, 4, 5), (statistical test used: Spearman R test at a significance level *p* = 0.05).

**Table 1 cancers-15-05013-t001:** Characteristics of primary antibodies.

Antibody	Clone	Dilution	Expression/Staining
anti-ERG	rabbit monoclonal antibody, Abcam, clone EPR3864(2), ab133264, Cambridge, UK	1:250	nuclear
anti-SPINK1	mouse monoclonal antibody, Abnova, clone 4D4, H00006690-M01, Taipei, Taiwan	1:100	cytoplasmic
anti-HOXB13	rabbit monoclonal antibody, Abcam, clone EPR17371, ab201682, Cambridge, UK	1:3000	nuclear
anti-TFF3	rabbit monoclonal antibody, Abcam, clone EPR3974, ab108599, Cambridge, UK	1:2000	cytoplasmic

**Table 2 cancers-15-05013-t002:** Scoring systems for immunohistochemical semi-quantitative assessment.

Antibody	Intensity of Labelling	Percentage of Positive Cells	IHC Score	Subgroup
ERG *	any intensity	any tumor cells displayed nuclear expression	positive	ERG positive
	absent	none	negative	ERG negative
SPINK1 **	any intensity	≥10%	positive	SPINK1 positive
	any intensity	<10%	negative	SPINK1 negative
HOXB13 ***	0 (negative)	0	I + P = 0–4	low HOXB13
	1 + (low)	1 (≤30%)		
	1 + (low)	2 (30%–70%)		
	1 + (low)	3 (>70%)		
	2 + (moderate)	1 (≤30%)		
	2 + (moderate)	2 (30%–70%)		
	3 + (strong)	1 (≤30%)		
	2 + (moderate)	3 (>70%)	I + P = 5–6	high HOXB13
	3 + (strong)	2 (30%–70%)		
TFF3 ****	0 (negative)	0—≤5%	I × P = 0–5	low TFF3
	1 + (low)	1 (6%–19%)		
	1 + (low)	2 (20%–49%)		
	1 + (low)	3 (>50%)		
	2 + (moderate)	1 (6%–19%)		
	2 + (moderate)	2 (20%–49%)		
	3 + (strong)	1 (6%–19%)		
	2 + (moderate)	3 (>50%)	I × P = 6–9	high TFF3
	3 + (strong)	2 (20%–49%)		
	3 + (strong)	3 (>50%)		

I = intensity of staining; P = percentage of positive cells, scoring system after * [33], ** [34], *** [35], **** [36].

**Table 3 cancers-15-05013-t003:** Clinico-pathological characteristics of the study group.

Clinico-Pathological Parameters	No of Cases	Percentage
Age at the Moment of Diagnosis		
50–59 years	7	9.72%
60–69 years	47	65.27%
>70 years	18	25%
Median age—66 ± 4.9 years/mean age—65.86 years		
Gleason score		
6 (3 + 3)	17	23.61%
7 (3 + 4)	38	52.77%
7 (4 + 3)	5	6.94%
8 (4 + 4)	5	6.94%
8 (5 + 3)	1	1.38%
9 (5 + 4)	6	8.33%
Prognostic grade group		
1	17	23.61%
2	38	52.77%
3	5	6.94%
4	6	8.33%
5	6	8.33%
Histological variant		
conventional ADK	72	100%
conventional ADK + foamy cell variant	9	12.5%
conventional ADK + mucinous variant	5	6.94%
conventional ADK + atrophic variant	1	0.72%
conventional ADK + microcystic variant	1	0.72%
Invasion aspects		
capsular invasion	66	91.66%
lymphovascular invasion	15	20.83%
perineural invasion	61	84.72%
pT stage		
T1	-	
T2	53	73.61%
T3	19	26.38%
T4	-	
Preoperative serum PSA levels		
≤4 ng/mL	2	2.77%
4.1–10 ng/mL	29	40.27%
10.1–20 ng/mL	23	31.94%
>20 ng/mL	18	25%
Postoperative serum PSA levels		
<0.2 ng/mL	37	51.38%
≥0.2 ng/mL	35	48.61%
Biochemical recurrence	35	48.61%

**Table 4 cancers-15-05013-t004:** Correlations between preoperative renal parameters in subsets of prostate acinar ADK defined by ERG/SPINK1 status.

Status	Preoperative Creatinine (mg/dL)		Preoperative Urea (mg/dL)		Preoperative eGFR (mL/min/1.73 m^2^)		Preoperative CKD Stage	
	Avg ± St. Dev.	*p*	Avg ± St. Dev.	*p*	Avg ± St. Dev.	*p*	Avg ± St. Dev.	*p*
E+S+ vs. E+S−	0.96 ± 0.17 vs. 0.96 ± 0.31	0.99	27.25 ± 10.81 vs. 31.40 ± 12.00	0.51	85.75 ± 16.78 vs. 88.97 ± 17.87	0.73	1.50 ± 0.58 vs. 1.42 ± 0.65	0.81
E+S+ vs. E−S+	0.96 ± 0.17 vs. 1.00 ± 0.34	0.79	27.25 ± 10.81 vs. 30.67 ± 8.35	0.54	85.75 ± 16.78 vs. 85.89 ± 21.73	0.99	1.50 ± 0.58 vs. 1.67 ± 0.87	0.73
E+S+ vs. E−S−	0.96 ± 0.17 vs. 1.01 ± 0.27	0.71	27.25 ± 10.81 vs. 31.05 ± 10.94	0.53	85.75 ± 16.78 vs. 84.35 ± 18.14	0.89	1.50 ± 0.58 vs. 1.61 ± 0.72	0.78
E+S− vs. E−S+	0.96 ± 0.31 vs. 1.00 ± 0.34	0.69	31.40 ± 12.00 vs. 30.67 ± 8.35	0.87	88.97 ± 17.87 vs. 85.89 ± 21.73	0.66	1.42 ± 0.65 vs. 1.67 ± 0.87	0.34
E+S− vs. E−S−	0.96 ± 0.31 vs. 1.01 ± 0.27	0.52	31.40 ± 12.00 vs. 31.05 ± 10.94	0.91	88.97 ± 18.87 vs. 84.35 ± 18.14	0.34	1.42 ± 0.65 vs. 1.61 ± 0.72	0.29
E−S+ vs. E−S−	1.00 ± 0.34 vs. 1.01 ± 0.27	0.97	30.67 ± 8.35 vs. 31.05 ± 10.94	0.93	85.89 ± 21.73 vs. 84.35 ± 18.14	0.84	1.67 ± 0.87 vs. 1.61 ± 0.72	0.85

Statistical test used: *t*-test at the critical significance level *p* = 0.05.

**Table 5 cancers-15-05013-t005:** Correlations between postoperative renal parameters in subsets of prostate acinar ADK defined by ERG/SPINK1 status.

Status	Postoperative Creatinine (mg/dL)		Postoperative Urea (mg/dL)		Postoperative eGFR (mL/min/1.73 m^2^)		Postoperative CKD Stage	
	Avg ± St. Dev.	*p*	Avg ± St. Dev.	*p*	Avg ± St. Dev.	*p*	Avg ± St. Dev.	*p*
E+S+ vs. E+S−	2.00 ± 0.00 vs. 0.92 ± 0.29	0.19	42.00 ± 5.83 vs. 35.10 ± 9.15	0.15	79.25 ± 9.18 vs. 90.92 ± 13.26	0.10	2.00 ± 0.00 vs. 1.36 ± 0.54	0.03
E+S+ vs. E−S+	2.00 ± 0.00 vs. 1.33 ± 0.50	0.02	42.00 ± 5.83 vs. 3.44 ± 10.25	0.06	79.25 ± 9.18 vs. 92.78 ± 17.75	0.18	2.00 ± 0.00 vs. 1.22 ± 0.67	0.04
E+S+ vs. E−S−	2.00 ± 0.00 vs. 1.48 ± 0.51	0.06	42.00 ± 5.83 vs. 31.83 ± 9.91	0.06	79.25 ± 9.18 vs. 86.4 ± 18.31	0.45	2.00 ± 0.00 vs. 1.57 ± 0.73	0.25
E+S− vs. E−S+	1.50 ± 0.51 vs. 1.33 ± 0.50	0.38	35.10 ± 9.15 vs. 30.44 ± 10.25	0.19	90.92 ± 13.26 vs. 92.78 ± 17.75	0.73	1.36 ± 0.54 vs. 1.22 ± 0.67	0.52
E+S− vs. E−S−	1.50 ± 0.51 vs. 1.48 ± 0.51	0.87	35.10 ± 9.15 vs. 31.83 ± 9.91	0.20	90.92 ± 13.26 vs. 86.43 ± 18.31	0.28	1.36 ± 0.54 vs. 1.57 ± 0.73	0.22
E−S+ vs. E−S−	0.92 ± 0.29 vs. 0.98 ± 0.26	0.61	30.44 ± 10.25 vs. 31.83 ± 9.91	0.73	92.78 ± 17.75 vs. 86.43 ± 18.31	0.38	1.22 ± 0.67 vs. 1.57 ± 0.73	0.23

Statistical test used: *t*-test at the critical significance level *p* = 0.05.

**Table 6 cancers-15-05013-t006:** Correlations between preoperative renal parameters in subsets of prostate acinar ADK defined by HOXB13/TFF3 status.

Status	Preoperative Creatinine (mg/dL)		Preoperative Urea (mg/dL)		Preoperative eGFR (mL/min/1.73 m^2^)		Preoperative CKD Stage	
	Avg ± St. Dev.	*p*	Avg ± St. Dev.	*p*	Avg ± St. Dev.	*p*	Avg ± St. Dev.	*p*
H+T+ vs. H+T−	0.98 ± 0.31 vs. 0.87 ± 0.12	0.54	31.32 ± 10.43 vs. 27.33 ± 11.15	0.54	86.73 ± 19.82 vs. 95.00 ± 8.72	0.49	1.55 ± 0.80 vs. 1.33 ± 0.58	0.66
H+T+ vs. H−T+	0.98 ± 0.31 vs. 1.20 ± 0.43	0.13	31.32 ± 10.43 vs. 32.67 ± 11.54	0.75	86.73 ± 19.82 vs. 74.00 ± 22.44	0.13	1.55 ± 0.80 vs. 2.00 ± 0.71	0.15
H+T+ vs. H−T−	0.98 ± 0.31 vs. 0.94 ± 0.23	0.53	31.32 ± 10.43 vs. 30.64 ± 11.60	0.82	86.73 ± 19.82 vs. 89.47 ± 15.58	0.55	1.55 ± 0.80 vs. 1.39 ± 0.59	0.41
H+T− vs. H−T+	0.87 ± 0.12 vs. 1.20 ± 0.43	0.23	27.33 ± 11.15 vs. 32.67 ± 11.54	0.50	95.00 ± 8.72 vs. 74.00 ± 22.44	0.15	1.33 ± 0.58 vs. 2.00 ± 0.71	0.17
H+T− vs. H−T−	0.87 ± 0.12 vs. 0.94 ± 0.23	0.60	27.33 ± 11.15 vs. 30.64 ± 11.60	0.64	95.00 ± 8.72 vs. 89.47 ± 15.58	0.55	1.33 ± 0.58 vs. 1.39 ± 0.59	0.86
H−T+ vs. H−T−	1.20 ± 0.43 vs. 0.94 ± 0.23	0.01	32.67 ± 11.54 vs. 30.64 ± 11.60	0.64	74.00 ± 22.44 vs. 89.47 ± 15.58	0.02	2.00 ± 0.71 vs. 1.39 ± 0.59	0.01

Statistical test used: *t*-test at the critical significance level *p* = 0.05.

**Table 7 cancers-15-05013-t007:** Correlations between postoperative renal parameters in subsets of prostate acinar ADK defined by HOXB13/TFF3 status.

Status	Postoperative Creatinine (mg/dL)		Postoperative Urea (mg/dL)		Postoperative eGFR (mL/min/1.73 m^2^)		Postoperative CKD Stage	
	Avg ± St. Dev.	*p*	Avg ± St. Dev.	*p*	Avg ± St. Dev.	*p*	Avg ± St. Dev.	*p*
H+T+ vs. H+T−	0.93 ± 0.23 vs. 0.95 ± 0.22	0.88	33.36 ± 11.57 vs. 40.67 ± 7.64	0.30	90.50 ± 15.49 vs. 86.33 ± 14.47	0.66	1.36 ± 0.66 vs. 1.67 ± 0.58	0.46
H+T+ vs. H−T+	0.93 ± 0.23 vs. 1.02 ± 0.16	0.34	33.36 ± 11.57 vs. 34.00 ± 10.42	0.89	90.50 ± 15.49 vs. 82.67 ± 14.74	0.21	1.36 ± 0.66 vs. 1.78 ± 0.44	0.09
H+T+ vs. H−T−	0.93 ± 0.23 vs. 0.93 ± 0.23	0.99	33.36 ± 11.57 vs. 33.57 ± 8.39	0.94	90.50 ± 15.49 vs. 89.97 ± 15.96	0.90	1.36 ± 0.66 vs. 1.39 ± 0.64	0.86
H+T− vs. H−T+	0.95 ± 0.22 vs. 1.02 ± 0.16	0.61	40.67 ± 7.64 vs. 34.00 ± 10.42	0.34	86.33 ± 14.47 vs. 82.67 ± 14.74	0.72	1.67 ± 0.58 vs. 1.78 ± 0.44	0.73
H+T− vs. H−T−	0.95 ± 0.22 vs. 0.93 ± 0.23	0.88	40.67 ± 7.64 vs. 33.57 ± 8.39	0.16	86.33 ± 14.47 vs. 89.97 ± 15.96	0.70	1.67 ± 0.58 vs. 1.39 ± 0.64	0.48
H−T+ vs. H−T−	1.02 ± 0.16 vs. 0.93 ± 0.23	0.31	34.00 ± 10.42 vs. 33.57 ± 8.39	0.89	82.67 ± 14.74 vs. 89.97 ± 15.96	0.22	1.78 ± 0.44 vs. 1.39 ± 0.64	0.10

Statistical test used: *t*-test at the critical significance level *p* = 0.05.

**Table 8 cancers-15-05013-t008:** Correlations between preoperative CKD stages and prognostic grade groups according to ERG/SPINK1 status.

Associations	E+S+ vs. E+S−		E+S+ vs. E−S+		E+S+ vs. E−S−		E+S− vs. E−S+		E+S− vs. E−S−		E−S+ vs. E−S−	
CKD stage 1—PGG 1	0	5	0	0	0	2	5	0	5	2	0	2
CKD stage 1—PGG 2	2	15	2	0	2	5	15	0	15	5	0	5
CKD stage 1—PGG 3	0	2	0	1	0	1	2	1	2	1	1	1
CKD stage 1—PGG 4	0	0	0	2	0	3	0	2	0	3	2	3
CKD stage 1—PGG 5	0	2	0	2	0	1	2	2	2	1	2	1
CKD stage 2—PGG 1	0	5	0	0	0	3	5	0	5	3	0	3
CKD stage 2—PGG 2	2	3	2	1	2	5	3	1	3	5	1	5
CKD stage 2—PGG 3	0	0	0	1	0	0	0	1	0	0	1	0
CKD stage 2—PGG 4	0	0	0	0	0	0	0	0	0	0	0	0
CKD stage 2—PGG 5	0	1	0	0	0	0	1	0	1	0	0	0
CKD stage 3—PGG 1	0	2	0	0	0	0	2	0	2	0	0	0
CKD stage 3—PGG 2	0	1	0	1	0	3	1	1	1	3	1	3
CKD stage 3—PGG 3	0	0	0	0	0	0	0	0	0	0	0	0
CKD stage 3—PGG 4	0	0	0	1	0	0	0	1	0	0	1	0
CKD stage 3—PGG 5	0	0	0	0	0	0	0	0	0	0	0	0
Total	4	36	4	9	4	23	33	9	36	23	9	23
χ^2^; *p*							4.54; 0.03					

PGG = prognostic grade group; statistical test used: Chi-square test at the critical significance level *p* = 0.05, with Yates correction.

**Table 9 cancers-15-05013-t009:** Correlations between preoperative CKD stages and prognostic grade groups according to HOXB13/TFF3 status.

Associations	H+T+ vs. H+T−		H+T+ vs. H−T+		H+T+ vs. H−T−		H+T− vs. H−T+		H+T− vs. H−T−		H−T+ vs. H−T−	
CKD stage 1—PGG 1	1	0	1	0	1	6	0	0	0	6	0	6
CKD stage 1—PGG 2	3	1	3	2	3	16	1	2	1	16	2	16
CKD stage 1—PGG 3	1	1	1	0	1	2	1	0	1	2	0	2
CKD stage 1—PGG 4	4	0	4	0	4	1	0	0	0	1	0	1
CKD stage 1—PGG 5	5	0	5	0	5	0	0	0	0	0	0	0
CKD stage 2—PGG 1	1	0	1	0	1	7	0	0	0	7	0	7
CKD stage 2—PGG 2	1	1	1	5	1	4	1	5	1	4	5	4
CKD stage 2—PGG 3	1	0	1	0	1	0	0	0	0	0	0	0
CKD stage 2—PGG 4	0	0	0	0	0	0	0	0	0	0	0	0
CKD stage 2—PGG 5	1	0	1	0	1	0	0	0	0	0	0	0
CKD stage 3—PGG 1	0	0	0	0	0	2	0	0	0	2	0	2
CKD stage 3—PGG 2	3	0	3	2	3	0	0	2	0	0	2	0
CKD stage 3—PGG 3	0	0	0	0	0	0	0	0	0	0	0	0
CKD stage 3—PGG 4	1	0	1	0	1	0	0	0	0	0	0	0
CKD stage 3—PGG 5	0	0	0	0	0	0	0	0	0	0	0	0
Total	22	3	22	9	22	38	3	9	3	38	9	38
χ^2^; *p*			7.63; 0.0057		3.99; 0.046 4.18; 0.036 6.68; 0.0097						6.84; 0.009 4.21; 0.04	

PGG = prognostic grade group; statistical test used: Chi-square test at the critical significance level *p* = 0.05, with Yates’ correction.

## Data Availability

The data used to support the findings of this study are available upon request to the authors.

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
