# Peer review of "Renal Function Parameters in Distinctive Molecular Subtypes of Prostate Cancer"

_cancers, 2023, doi:10.3390/cancers15205013_

Round 1
Reviewer 1 Report
A research manuscript titled, "Renal function parameters in distinctive molecular subtypes of prostate cancer" by Timofte et al., described the implications of various parameters involved in prostate cancer. The manuscript was well-written, well-organized and focused on SPINK1, ERG, HOXB13, and TFF3 markers. FOllowing minor corrections are suggested before the acceptance:-
1. In the 1st and 2nd para- little plagiarism detected need to be corrected accordingly.
2. A good number of patients, 72 selected have cancer plus CKD, Authors need to clarify the age group variation, and androgenic hormonal conditions impact on the cancer-associated CKD.
Reviewer 2 Report
Timofte demonstrated renal function parameters in distinctive molecular subtypes of prostate cancer. The study underscores the variability of renal function parameters in different molecular subtypes, offering potential insights into patient prognosis. The manuscript is very interesting, but there are several issues that should be addressed.
- References should be written at the end of the sentence instead of the middle, for example reference number 7. Authors should carefully revise the references.
- Lines 71, 72, 73: The authors should provide the references for these information's.
- The introduction part lacks emphasis on the background. I recommend a revision of the introduction and add a sufficient background.
- Inclusion and Exclusion Criteria and Experimental Design should be provided as in a diagram under the materials and methods section.
- I recommend representing the clinico-pathological characteristics in a table.
- The study limitations should be provided.
- At the bottom of the table, you must place which statistical test you use.
- I recommend placing more emphasis on the novelty of the article and to revise conclusions.
- There are many grammar mistakes and typo-errors. I highly recommend the careful revision of the manuscript.
Moderate editing of English language required
Reviewer 3 Report
Title:
Renal function parameters in distinctive molecular subtypes of prostate cancer by
Andrei Daniel Timofte, et al.
Description: (adapted from Abstract)
Prostate cancer is a prevalent malignancy in male patients, having diverse clinical outcomes. This study aimed to investigate the relationship between renal function parameters and distinctive molecular subtypes of prostate adenocarcinomas, defined by the immunoexpression of SPINK1, ERG, HOXB13, and TFF3 markers. The study group comprised 72 patients with prostate cancer and associated chronic kidney disease (CKD) who underwent radical prostatectomy. Histopathological, molecular, and renal parameters were analyzed. Patients were categorized based on ERG/SPINK1 and HOXB13/TFF3 status, and correlations with renal function and prognostic grade groups were assessed. The ERG+/SPINK1+ subgroup exhibited significantly higher postoperative CKD stage and serum creatinine levels compared to the ERG+/SPINK1- subgroup. This suggests an intricate relationship between SPINK1 overexpression and renal function dynamics. The HOXB13-/TFF3+ subgroup displayed higher preoperative serum creatinine levels and CKD stage than the HOXB13-/TFF3- subgroup, aligning with TFF3's potential role in renal function. This research underscores the variability of renal function parameters in different molecular subtypes, offering potential insights into patient prognosis.
Strengths:
The manuscript describes an interesting observation that certain renal biomarkers correlate with the prognosis of prostate cancer. The data overall supports the conclusion and may point a direction for prostate cancer therapy.
Weaknesses: Some suggestions and concerns are listed below.
1. Result section: the 1st paragraph, i.e., the first 3 sentences (lines 150 - 152), are redundant and can be removed.
2. Results section: The 2nd paragraph (lines 153 - 154) can be combined into the Material section.
3. Results section: Fig. 2 is not clear. Figure legend may be provided with explanation of its analysis.
4. The discussion should be shortened for its succinct.
Reviewer 4 Report
This manuscript highlights a clinical observation that connects prostate cancer development and treatment with chronic kidney disease. The authors try to integrate the molecular subtypes with renal function to predict the aggressiveness of prostate acinar adenocarcinoma, which might be interesting for the overall research field. However, there are a few concerns that need to be addressed.
Major:
1. Is there any rationale for choosing these four markers (ERG, SPINK1, HOXB13 and TFF3)?
2. In the result section, it is unnecessary to divide the comparison results of the same genomic subtype into two tables, such as Tables 3 and 4 or 5 and 6.
3. Putting a large bulk of text about the characteristics of staining patterns in the results section didn't help the authors prove this manuscript's purpose because it is separate from the results. This part should belong to either the method section or the supplementary material.
Minor:
1. A paragraph of the previous reviewer's comment was copied into the main text (line 150).
2. For Figure 1, adding the labels of different markers should be better for presentation purposes.
Reviewer 5 Report
Dear Authors,
Despite an interesting topic I have some objections for this manuscript:
1) Introduction. Please, move the last paragraph - Line 77-84 to the Discussion. That what you mentioned there should be said, but in other place;
2) Materials and methods.
-Clarify, please, the inclusion/exclusion criteria for the patients.
-Control group is mandatory, but here is absent. Please, include the control! And give also its detailed description.
-Describe, please, the tissue collection and fixation procedure, detailed IMH protocol and give reference for it. Add here also the haematoxylin and eosin staining as review slides are also mandatory for the histopathology!
- add, please, the antibodies negative and positive control description!
3) Results.
-Move Line 150-153 to the M+M section.
-Please, add review microphotos and their description.
- please, add the control microphotos for control and each specific antibody /cancer type used.
4) Discussion.
- add, please Limitations paragraph at the end of this section.
5) Conclusions. Make them more precise by giving the data shortly on which the main conclusion is developed. Remove, please, the 2nd sentence what is really extra and useless.
Round 2
Reviewer 2 Report
The authors addressed all my concerns. I recommend the acceptance of the manuscript.
Reviewer 4 Report
The authors have addressed all my previous concerns. There are no more concerns regarding this manuscript.
Reviewer 5 Report
Dear Authors,
Well, You indicated that "this type of study does not require a control group", this is not exactly so (because some controls in specific circumstances can show some exceptions what could be similar to your results and what you will do then!?) This view, sorry, not to have controls during the comparison a pathological data, is rather "old fashion" style for the scientific papers than objectivity! But I remember this style, too! And thus I "close" my eyes for it...
Commonly the huge work is done, thank you and I will advice your manuscript for publication! (Thanks for the pictures, very nice!)